# The TELE-DD Randomised Controlled Trial on Treatment Adherence in Patients with Type 2 Diabetes and Comorbid Depression: Clinical Outcomes after 18-Month Follow-Up

**DOI:** 10.3390/ijerph20010328

**Published:** 2022-12-25

**Authors:** María Luisa Lozano del Hoyo, María Teresa Fernandez Rodrigo, Fernando Urcola-Pardo, Alicia Monreal-Bartolomé, Diana Cecilia Gracia Ruiz, Mercedes Gómez Borao, Ana Belén Artigas Alcázar, José Pedro Martínez Casbas, Alexandra Aceituno Casas, María Teresa Andaluz Funcia, Juan Francisco Roy Delgado

**Affiliations:** 1Las Fuentes Norte Health Centre, Servicio Aragonés de Salud (SALUD), 50002 Zaragoza, Spain; 2Department of Physiatry and Nursing, Faculty of Health Sciences, University of Zaragoza, 50009 Zaragoza, Spain; 3Water and Environmental Health Research Group (DGA-B43-20R), 50009 Aragón, Spain; 4Aragon Institute for Health Research (IIS Aragon), 50009 Zaragoza, Spain; 5Primary Care Prevention and Health Promotion Research Network, RedIAPP, 28029 Madrid, Spain; 6Sagasta Health Center, Servicio Aragonés de Salud (SALUD), 50006 Zaragoza, Spain; 7University Hospital Miguel Servet, Servicio Aragonés de Salud (SALUD), University of Zaragoza, 50009 Zaragoza, Spain; 8San Pablo Health Center, Servicio Aragonés de Salud (SALUD), 50003 Zaragoza, Spain; 9Faculty of Health Sciences, Universidad San Jorge, Villanueva de Gállego, 50830 Zaragoza, Spain

**Keywords:** diabetes mellitus type 2, depressive disorder, diabetes distress, healthy lifestyle, motivational interviewing, primary care nursing, randomised controlled trial, self-management, telephonic intervention, treatment adherence

## Abstract

Clinical depression is associated with poorer adherence to hypoglycaemic medication in patients with diabetes mellitus, leading to poorer glycaemic control, diabetes management, and increased complications. The main aim of the TELE-DD trial was to demonstrate the efficacy of a proactive and psychoeducational telephonic intervention based on motivational interviewing and collaborative care to reduce nonadherence and improve prognosis in individuals with diabetes mellitus and concurrent depression. Design: The TELE-DD project is a three-phased prospective study including a nested randomised controlled trial. Methods: The baseline cohort included the entire population of adult patients diagnosed with type 2 diabetes and concurrent depression. A randomised controlled trial was conducted in a selection of patients from the baseline cohort, distributed into a control group (n = 192) and an intervention group (n = 192). Monthly telephonic interventions delivered by specifically trained research nurses were centred on a psychoeducational individualised monitoring protocol including motivational interviewing and collaborative care strategies. Clinical and patient-centred data were systematically collected during an 18-month follow-up including HbA1c, Patient Health Questionnaire, and the Diabetes Distress Scale. Results: During the trial, 18-month follow-up HbA1C levels significantly (*p* < 0.001) decreased in the intervention group at every follow-up from an average of 8.72 (SD:1.49) to 7.03 (SD:1.09), but slightly increased in the control group from 8.65 (SD:1.40) to 8.84 (SD:1.38). Similar positive results were obtained in depression severity and diabetes distress, LDL-cholesterol, and systolic and diastolic blood pressure, but only at the 18-month follow-up in body mass index reduction. Conclusions: This is the first trial to concurrently decrease biological and psychological outcomes with a monthly brief telephonic intervention, pointing out that a combined biopsychosocial intervention and collaborative care strategy is essential for current world health challenges. ClinicalTrials.gov Identifier: NCT04097483. Patient or Public Contribution: Diabetic patients not belonging to the TELE-DD population or trial sample were consulted during the study design to review and guarantee the clarity and understanding of the trial psychoeducational materials.

## 1. Introduction

The prevalence of diabetes, deaths attributable to diabetes, and health expenditure due to diabetes continue to increase worldwide with significant social, financial, and health systems implications [1] The World Health Organisation (WHO), in a new report on the monitoring of noncommunicable diseases, Type 2 diabetes (T2D) among them, exposed that there is a need to intensify control measures to reduce further complications and the number of premature deaths. The objective of T2D treatment is to prevent or delay complications and maintain quality of life since cardiovascular disease is increased in people with diabetes, being its leading cause of death [2]. Projects such as The Diabetes Control and Complications Trial Research Group (DCCT), the UK Prospective Diabetes Study (UKPDS), and high-quality clinical trials have shown that it is possible to lower glycated haemoglobin (HbA1c) levels to 7%, especially in the early stages of diagnosis, and consequently, long-term complications can be substantially reduced [3]. Several studies have shown that antihypertensive and blood lipid-lowering medications can reduce cardiovascular risk in people with diabetes, with the same effectiveness as in the non-diabetic population [4] The WHO recognises that there is evidence that simple lifestyle measures are effective in preventing T2D or delaying its onset [5]. Medications are one of the main therapeutic resources for health care; However, their benefits can be altered by a lack of compliance or therapeutic adherence. Problems related to non-compliance with treatment are observed in all situations in which the treatment must be administered by the patient himself, regardless of the type of disease [6].

Diabetes mellitus (T2D) is a complex pathology that poses a daily challenge for people who suffer from it and when it is associated with clinical depression, the complexity increases exponentially [7]. Both diseases are bidirectionally related [8]; it is estimated that between 18% and 35% of the population with diabetes mellitus are diagnosed and treated for depression [9]. This implies a lower quality of life and higher mortality rates, given the functional deterioration due to the lack of self-management and development of comorbidities as well as having social and economic consequences. Most studies demonstrate a strong association between non-adherence and lack of glycaemic control [10]; González et al. [11], in a prior meta-analysis, demonstrated that depression is associated with lower adherence to hypoglycaemic medication in patients with T2D, which leads to worse glycaemic control and poor management, and consequently, an increase in complications and premature death [12], with significant prognostic and psychosomatic implications [13]. Psychological distress in patients with diabetes (DD) is a syndromic level comorbidity that manifests with multiple feelings such as worry, conflict, frustration, and discouragement while being closely related to the symptomatology of depression [14]. In these circumstances, problem-solving skills and self-care efficacy are impaired, leading to poor glycaemic control, elevated lipid levels, and a worse prognosis. Nicolucci et al. [15] suggest that affective well-being is the most negatively worsened domain by diabetes, second only to physical status and health. 

### Background

Given the importance of treatment adherence (TA) in the evolution of chronic disease, and diabetes being one of the diseases with the lowest level of self-care, we consider that achieving adherence to treatment in people with T2D and DD through a controlled and structured intervention, by nursing professionals in the context of primary care could be successful in reducing both the physical and psychological clinical outcomes [16]. The population context in which the trial was developed comes from the preliminary phase I study. These were adult patients (+21 years old), diagnosed with type 2 diabetes (T2D) and concurrent depression and pharmacological treatment (n = 3601). This population was from Region Zaragoza II from the Aragonese Health System (SALUD), in the city and province of Zaragoza, Aragon (Spain). About one-third (35.4%) of TELE-DD patients were non-adherent to either or both pharmacological treatments (n = 1274) [17]. Thus, a better TA is related to better glycaemic stability and prognosis, and a lower health care use; for this reason, any intervention committed with effectiveness must be fully integrated with primary care (PC) usual clinical practice and care, where a higher interaction between patient and health professionals is found, and where the efficacy of patient care is optimal [18]. Therefore, the development of integrative strategies with patients, primary care specialists (PCS), nurses, psychologists, and caregivers to promote TA can be fundamental to achieving successful and sustainable self-care and prognosis in T2D patients with comorbid depression. TA is considered a key factor to improve health outcomes, and the most used and tested methods to control and improve TA are psychological or educational interventions and telemedicine/telephonic monitoring. The choice of this approach is not random, as a meta-analysis of ninety-four studies by Horne et al. [19] placed diabetes mellitus among the diseases with the lowest levels of self-care, their findings showing that there is sufficient evidence that links a better adherence to a greater perception of the need for treatment and less concerns about adverse effects.

Our research questions were as follows: Can an organised nursing intervention based on an interpersonal and direct relationship with the patient over the telephone with diabetes and comorbid depression, be effective in promoting adherence and self-care? Will this intervention improve clinical outcomes such as blood glucose, depressive symptoms, and diabetes distress in these patients?

Aims: The main aim of the TELE-DD trial was to demonstrate the efficacy of a joint psychoeducational intervention delivered by nurses via monthly telephone calls to increase TA and further clinical outcomes in individuals with T2D and concurrent depression, in a random stratified sample of patients. We propose conducting a telephone intervention to analyse the efficacy of an innovative and integrative approach including motivational interviewing, cognitive-behavioural therapy for anxiety and depression (CBT-AD) strategies, collaborative care and additional education, nutrition counselling and follow-up, with the encouragement of self-management and physical activity, social support, and other minor therapeutical objectives. These approaches, which have proven positive in treatment adherence, will be directly translatable to PC services. 

## 2. Materials and Methods

Design: The current report includes the TELE-DD project Phase II, a randomised controlled trial (RCT) nested in a population-based cohort study (TELE-DD Phases I and III), [17] in the city of Zaragoza (Spain). The Phase II RCT was based on a motivational nurse-led proactive monthly intervention by telephone, centred in a psychoeducational personalised monitoring protocol. In the prior TELE-DD Phase I report [17], findings from the baseline cohort data showed an average patient profile of an elder woman with low treatment adherence, pre-obesity or Stage-I obesity, elevated LDL-cholesterol, and high HbA1c, treated with several drugs (polypharmacy), and frequent visits to primary care services. The majority lived in poor urban areas, 12% of them were living alone, and showed to be moderately disabled (average dependency rate of 52%); these relevant population data (N = 3601) guided the RCT design. A two-arm parallel-group single-blind pragmatic RCT design was implemented to objectively compare the effect of psychoeducational telephonic monitoring of diabetic patients with comorbid depression in real-world health system PC services, instead of demonstrating causal explanations for traditional outcomes. The RCT was implemented under an intention-to-treat design. All randomised patients were followed up for 18 months and completed follow-up 6, 12, and 18-month assessments until the planned conclusion of the TELE-DD RCT. 

Tele-DD Phase II RCT sample size, participants, and power calculations. The current report includes the TELE-DD project Phase II, a randomised controlled trial (RCT) nested in a population-based cohort study (Phase I) [17], that included the entire population of adult patients (21+ years) diagnosed with type 2 diabetes (T2D) and concurrent depression (N = 3601), not adherent to any treatment (n = 1274). Phase II of TELE-DD was conducted in a randomised selection of patients from the baseline Phase I cohort (Figure 1). 

The RCT sample was stratified by age, sex, and health centre in order to ensure representativity from the patient population (see Phase I report design). Participant recruitment and implementation of the TELE-DD RCT protocol took place from January to December 2017 at the 23 primary care centres located in Zaragoza and several rural neighbouring districts and towns belonging to Aragon’s health service SALUD Sector II. Sample size was calculated through the finite population formula, considering that the population not adhering to any treatment was 1274 people (Figure 1), representing 35% of the reference diabetic population with comorbid depression, receiving pharmacological treatment, and belonging to Sector II of the SALUD Health Service; based on the reference population data, we estimated a sample size of 360 patients. Additionally, the minimal clinically significant reduction at the end of the 18-months RCT follow-up was set to an IFCC HbA1c 10.9 mmol/mol (NGSP HbA1c 1%) difference in the TIG compared with TAU, considering that HbA1c may be reduced by TAU in 2.2 mmol/mol or NGSP HbA1c of 0.2%. We assumed a 20% dropout, which leads to a sample of 360 patients in both RCT arms to reach 80% power at a 5% two-sided a-level with 10 × 2 research nurses per arm. With an estimation of two research nurses per arm to potentially drop out during the follow-up due to health centre or region relocation, sick leave, or any other reason, we recruited 12 × 2 research nurses, for a minimum required total sample size of n = 432 patients from the 23 health centres that participated in the TELE-DD study, a number set to guarantee a minimum and safe sample size and avoid sample attrition. 

The correspondent number of patients was randomly assigned to the control group, which allowed a comparison of the effect of the intervention on the intervention group. Once the project began, no new patients were added to the sample. The intervention group (IG) consisted of 225 patients and the control group (CG) included 203 patients. The allocation of the number of participants per health centre was proportional to the number of users at each health centre in the reference group (non-adherents to T2D and DD medication). The participants who completed the entire study period were 198 in the control group and 198 in the intervention group, whose data were finally analysed. The causes of dropouts were death, inclusion in a nursing home, and change of residence (Figure 1). 

Data collection: Phase I baseline cohort: Patient’s eligibility, diagnostic criteria, and inclusion/exclusion criteria. Eligible for inclusion in the TELE-DD baseline cohort were all adult individuals (21+) with concurrent T2D and depression diagnosis registered in the SALUD clinical computerised system before 1 January 2016, according to the Central Health Services-Electronic Medical Records (CHS-EMR), a systematic registration tool that records the clinical and demographic data of every patient including laboratory data, primary care, or specialist intervention; if empty fields were found in the CHS-EMR platform, the TELE-DD research nurses investigated any health service records and filled in the missing values. Collected and registered primary and secondary outcomes and other covariates here described were and will be measured, classified, and analysed under the same criteria across the TELE-DD population-based cohort, nested RCT, and further studies (Phases I to III, while the third phase will consist of a 5-year prognostic and cost-effectiveness study). The diagnostic system used in CHS-EMR is the International Classification of Primary Care—2nd Edition (ICPC-2) codes T90 (type 2 diabetes) and P76 (depressive disorder). Every diagnosis of diabetes, depression, and other illnesses were confirmed by the corresponding primary care specialist; the ICPC-2 system has the highest specificity due to its conversion structure with the International Classification of Diseases (ICD-10) [20]. Additionally, patients were included if both T2D and depression diagnoses concurrently lasted at least one year. Exclusion criteria included: dementia, AD, or severe psychiatric or cognitive illness; patients treated on private health insurance that may potentially skew the RCT results; absence of drug prescription for T2D or clinical depression; unable to locate the patient due to address or residence change out of SALUD Zaragoza Region II. 

Phase II randomised controlled trial: Patient’s eligibility and inclusion/exclusion criteria. In consonance with the RCT design and eligibility criteria, the Tele-DD RCT reflected a balance between the usual strict requirements in T2D and depression trials and those oriented to pragmatic and future translatable clinical objectives. In addition to the previously mentioned inclusion and exclusion criteria for the baseline cohort study, new criteria were specifically defined and added for the RCT patient’s eligibility. Inclusion criteria: willingness to adhere to the nursing intervention protocol including self-monitoring of blood glucose; able to speak Spanish; signing of the RCT informed consent before any study inclusion. Exclusion criteria: PCS unwillingness to communicate information about the study and the participant’s data; no PCS or inability to identify a reference one, as PCS and PC nurses provided treatment as usual (TAU) by the time of final run in; hearing impairment and/or inability to understand and speak the Spanish language; limited or no access to a telephone at the patient’s home or permanent residence, or to a mobile phone. Participants who met the inclusion/exclusion criteria received a code from the screening research nurses to maintain the participants’ confidentiality. 

Randomisation and allocation. Only patients not showing TA for both T2D and depressive disorders were randomly assigned to one of the RCT two arms: the intervention group (IG) that included the telephonic structured research interview, collaborative care, and the psychoeducational intervention plus TAU, or the control group with TAU only. In addition, the TAU-only control group was also assessed for the same covariates and outcome measurements as the TIG patients at the baseline and during the three follow-up assessments. The randomisation process was performed by an independent public-health researcher who was unaware of the study characteristics. TELE-DD RCT participants were randomly allocated by using a computer-generated random number sequence through a simple allocation strategy and a 1:1 rate. Patients agreed to participate before random allocation while being blind to the allocation process. The researcher who administered the baseline cohort data collection, management, and analyses was blinded to the treatment group to which every patient belonged. This investigator was different from those of the research nurses who administered the intervention and collected measures throughout the RCT study follow-up assessments. The PCS was blind, but the research nurses carrying out the telephone protocol were not. Due to ethical reasons, the RCT participants were not blind to the treatment condition once allocated, the trial being unmasked. Evaluations of the primary outcome measures and additional secondary outcomes were obtained from CHS-EMR and through the TELE-DD structured interview at the baseline and in every follow-up (6, 12, and 18 months from Phase II RCT baseline assessment). The TELE-DD RCT study followed the Consolidated Standards of Reporting Trials (CONSORT) guidelines for reporting social and psychological intervention trials [21]. Trial unique registration: clinicaltrials.gov Identifier: NCT04097483.

Measures validity and reliability. Treatment Adherence (phase I). We measured TA through the medication possession ratio (MPR), a standard measure of TA defined as the number of drug units dispensed divided by the number of drug units expected for a specific observational period. We calculated TA as the proportion of the number of days with the treatment provided during the intended period of treatment: 100 × ∑ (days supplied)/365. The number of days on which treatment was given was estimated according to the number of drug units dispensed during the observation year, assuming in our case that the treatment dosage was one drug unit per day during the year before the baseline assessment [22]. Once TA was computed, a dichotomous variable was created as TA (Yes/No) considering a cut off ratio of MPR ≥ 80% [22,23,24].

Metabolic outcomes (Phase I; Phase II). Glycosylated Haemoglobin (HbA1c) levels were obtained from the CHS-EMR and classified according to the American Diabetes Association (ADA) standards of medical care on the classification and diagnosis of diabetes [25]. Previous studies [26] found that patients with an HbA1c greater than or equal to 7% had a 166% increased risk for all-cause mortality (risk ratio = 2.66 95% CI: 1.16 to 6) and an increased risk for cardiovascular mortality of 3.50 (95% CI: 1.09 to 11.23). LDL-Cholesterol: LDL-cholesterol was measured under the reference data and recommendations from the National Cholesterol Education Program Adult Treatment Panel III (2001): <100 mg/dL optimum, 100–129 mg/dL normal or close to an optimal level, 130–159 mg/dL normal-high, 160–189 mg/dL high, >190 mg/dL very high. The International Atherosclerosis Society [27] and the European Society of Cardiology recommend low-LDL cholesterol levels of 100 mg/dL in people with diabetes and without CVD. In addition, a LDL-cholesterol goal of less than 70 mg/dL should be reached in T2D patients with high cardiovascular distress (presence of CVD or associated risk factors) [28]. Psychological outcomes: Patient’s psychological and emotional state. The Patient’s Health Questionnaire (PHQ-9) was used to assess the patients’ affective states. PHQ-9 is widely used in patients with chronic diseases including diabetes [29] and was primarily designed to screen for the presence and severity of depression in PC settings. In the general population, sensitivity (77.5%; 61.5–89.2) and specificity (86.7%; 83.0–89.9) scores for the identification of a major depressive episode were found [30]. This evaluation was not a diagnosis of depression in the patients. They were recruited because they had a clinical depression diagnosis under the ICD-10 criteria (see above). Additionally, specific T2D-related distress was measured with the Diabetes Distress Scale (DDS) [31]. The DDS has been developed specifically for the diabetic population, with the participation in its development by diabetic patients, diabetes specialist nurses, nutritionists, diabetologists, and psychologists. Prior studies have shown that DDS presents high sensitivity (95%) and specificity (85%) with good reliability (α = 0.93) for investigation and clinical work [32]. The patients’ behaviour, attitudes, or activities detected during the intervention that could negatively influence their optimal control were notified to the PCS or nursing professionals who applied the standard diabetes treatment protocol in every individual case. 

Covariates (Phase II). Patient’s clinical history: At the baseline, cohort assessment data were collected from diabetes mellitus diagnosis time and also the time of the first clinical depression confirmed diagnosis according to CHS-EMR, and process of care indicators including the number of yearly visits to PCS and nurses, detailed drug prescription, and BMI. Previous reports indicated a significant association between hypertension and depression in the elderly population [33] and showed that a diastolic blood pressure (DBP) difference of 5 mmHg significantly reduced the mortality from diabetes [34]. Low physical activity, overweight/obesity, smoking behaviour, and HTA have recently been identified as unique and shared risk factors for T2D and depression [35]. 

Diabetes mellitus and comorbid depression self-management strategy, collaborative care approach, and psychoeducational nurse-led intervention protocol and rationale. The TELE-DD Project was presented to all the PCS and nursing professionals of the 23 participating health centres from Region II of Zaragoza (SALUD) to arrange and further guarantee the achievement of all the objectives of the study, with a special mention to collaborative care strategies. The intervention followed the ADA 2015 T2D self-management education and support consensus statement and guidelines [36] and consisted of a monthly structured telephone call with an average duration of 30 min per call. Research nurses received highly specific training and standardisation on the study protocol and integrative strategies. In addition, they collaborated with physicians through previously established collaborative care strategies to achieve an individualised and integrated care management through education and guideline-based treatment recommendations, maximising strategies for an optimum TA as well as the patients’ commitment to polypharmacy and disease self-management, while promoting a healthy lifestyle. In summary, core nurse-led patient self-management strategies included: (1) Refilling pillboxes weekly, checking and correct dose use; (2) Using a treatment scheduling sheet to mark the date and time of drug intake, glucose monitoring, and other treatment tasks; (3) Associating while synchronising daily treatment tasks including nutrition and physical activities as part of another routine (i.e., reading/watching the news in the morning or before/after every meal); (4) Using stickers or other reminders for self-management treatment tasks; (5) Any other strategy the research nurse believed that was useful for every single patient’s full treatment compliance including information technologies or anything else that would help the patient to reach a 100% integrative TA and appropriate self-care management. Additionally, physical, psychological, and social advice and support strategies were addressed by the research nurses by using specific and empathic language and communication skills. The main elements of the psychological and behavioural side of the intervention (CBT-AD) included: (1) Adherence to treatment, dose and schedule review, self-management, and importance of the health professional–patient relationship; (2) Education and intervention in healthy life habits and behaviours as a fundamental part of their treatment (food and exercise); (3) Addressing their needs and concerns; (4) Emotional distress management skills and cognitive assessment for problem-solving in any issue related to T2D and depression treatment; (5) Use of motivational interviewing as a way of solving problems of self-management and emotional distress; (6) Cause behavioural changes and help patients to explore and solve self-control problems; (7) Improvement in the patients’ QOL and well-being as a rule.

Statistical analysis for clinical efficacy: Descriptive statistics of the included variables (mean and 95% confidence interval for normally distributed quantitative variables; and median and interquartile range for abnormally distributed quantitative variables) were performed with the same strategy as our prior TELE-DD Phase I report [17,37]. All comparable study variables as clinical and metabolic indicators were measured at the baseline and the 6, 12, and 18-month follow-up assessments. For each variable, the treatment group was compared separately within each subgroup with a test of homogeneity between strata. Clinical efficacy was tested through ‘head-to head’ comparison between the two groups (TAU and TIG). After RCT completion, statistical analyses followed the intention-to-treat principle to compare the randomly assigned TIG and TAU groups with all the available data from every subject included in the study, with no exception. To confirm the main hypothesis, all variables were compared (t0-tk) using an analysis of variance (ANOVA) and post hoc tests or Kruskal–Wallis non-parametric tests [37].

## 3. Results

Participants were recruited from January to December 2017. A total of 1274 subjects with type-2 diabetes and comorbid depression from the TELE-DD Phase I cohort study were approached for the Phase II trial eligibility, of which 428 (34%) eligible participants were recruited in the study after the inclusion and exclusion criteria were applied. These 428 patients were stratified by sex, age, and health centre, and randomised by using a computer-generated random number sequence for the intervention or control group random allocation. During the 18-month follow-up, twelve (6%) participants from the control group and thirty-four (15%) participants from the intervention group dropped out for the reasons listed in Figure 1. The overall attrition rate was 11%, and 382 participants completed all of the data collection.

### 3.1. Characteristics of the RCT Sample

There were no differences in the mean values of the sociodemographic characteristics between the control group and the intervention group; the mean age was 70 years, and the proportion of women was much higher in both groups than the proportion of men. Time to disease diagnosis and the mean visits to the doctor or nurse were also similar in both groups. These data indicate that the total sample population was very uniform (Table 1).

### 3.2. Metabolic Outcomes

The trial primary outcome was glycated haemoglobin (HbA1c) due to its proven causality on T2D complications. Comparing the values at month 0 and month 18, the mean HbA1c value in the CG increased slightly (8.65 at month 0 to 8.84 at month 18), and in the IG, it was clearly decreased (8.72 at month 0 to 7.03 at month 18) in all three cases; *p* < 0.01, including the interaction of both factors (Table 2) (Figure 2). 

Secondary outcomes: included LDL-C and BMI. LDL-cholesterol: Mean values increased slightly over the trial 18-months follow-up in the CG (117.07 mg/dL month 0; 125.87 mg/dL month 18); the IG values decreased at the end of the trial but slightly increased between months 12 and 18. Test values were significant (*p* < 0.001) in the three cases, both in the comparison by time and between groups as well as in the interaction of both factors (Table 2). Mean BMI values of both CG and IG groups at month zero were similar (around 30), while in the CG, there was a slight increase (30.49 at month 0; 30.78 at month 18; difference: plus 0.29), the mean BMI in the IG decreased slightly and progressively during follow-up (30.42 at month 0; 30.05 at month 18; difference: minus 0.37). BMI was the variable that was proven to remain the most constant over time in the IG. Mean systolic blood pressure in the CG increased by almost thirteen points (134.41 mmHg month 0; 147.29 mmHg month 18), but only slightly in the IG (difference: 0.31 points). The evolution of the mean diastolic blood pressure values of both groups at different times had similar behaviour, decreasing slightly, although more so in the IG. In the CG, the total difference between the initial and final time was 2.61 points, while in the IG, it was 5.24 points. In the respective statistical tests, significant differences were obtained in both the comparison by time and by group as well as in the interaction *p* < 0.001 (in the latter, the significant differences occurred between the groups from month 12 onward) (Table 2) (Figure 3 and Figure 4).

Psychological outcomes: Table 3 shows the evolution of the trial results regarding psychological outcomes. At month 0, patients who indicated clinical signs and symptoms compatible with major depression (PHQ-9 cut-off criteria) were 28% of the population, 34% with minor depression, and 38% had no symptoms of depression. 

A total of 73.5% patients felt tired or had low energy, 56.5% had problems sleeping, staying awake, or sleeping too much, and 27.6% of those patients thought that they “would better be dead” or had wished they had harmed themselves in some way. A comparison of the PHQ-9 results over the 18-months follow-up of CG and IG initially showed (at month 0) great similarity between the groups, with symptoms related to the “No depression” (CG 36.9%; IG 3.8%) and “Minor depression” classification being more frequent (CG 48.3%; IG 48%). At month 6 in the CG, the most reflected symptoms were those compatible with “Minor depression” (54.5%), and in the IG, the symptom category of “No depression” (52%), and “Major depression” were very similar in both groups (CG 12.6%; IG 9.1%). At month 12, the CG continued to show a total score in the same range, and “Major depression” did not decrease in proportion, while in the IG, the “Major depression” category decreased notably in proportion, and at month 18, the “No depression” category was the most frequent in IG patients (62.3%). Statistical tests showed no significant differences between groups at month 0, but positive significant results from 6 to 18-months from the trial start (*p* ≤ 0.001) (Table 3). The mean total DDS questionnaire scores in the trial sample (CG and GI) before the start of the trial were 1.48 ± 0.4. The mean values of the DDS four domains were analysed individually: emotional burden obtained a mean value of 1.40 ± 0.41, interpersonal distress 1.51 ± 0.41, distress 1.61 ± 0.54, and medical distress 1.39 ± 0.53. Comparative data of the DDS total score between the two groups showed that in the CG, the mean remained almost unchanged over time (1.50 at month 0;1.48 at month 18), while the DDS total mean score of the IG clearly decreased (1.48 at month 0; 1.18 at month 18). Although the initial values of the test indicate a state of the patient with little or no distress, these values continued to decrease in the IG compared to the CG, which remained at similar values to the trial baseline (month 0).

## 4. Discussion

The chronic nature of T2D requires patients to take complex self-management and daily management throughout their lives since diagnosis. These daily management strategies include proper nutrition, physical exercise, medication, and blood glucose control to maintain a good quality of life. As diabetes is linked with clinical depression, there is an added difficulty in demanding self-care. Non-adherence to medication and the prescribed lifestyle activities leads to a demonstrated risk for the development of short and long-term complications. The reference population cohort (Phase I) of the trial was made up of non-adherents to pharmacological treatment and was characterised by a higher proportion of women, a mean age of 71.4 years, and more years of diagnosis of T2D than MDD. The clinical parameters of HbA1c, LDL-cholesterol, blood pressure, and BMI were above the optimal values at the start of the study [17]. Another characteristic of this population that could be a contributing factor to nonadherence is the high mean number of drugs simultaneously prescribed, which was 6.4, like those reported by Patel et al. (2017) [38]. Polymedication affects adherence and, specifically in the elderly population, is a cause of self-management mistake, and forgetfulness. 

Health care focused on patient care equity, while taking into account their opinions, issues, or particular characteristics, and the establishment of a professional patient–nurse relationship, allows for full involvement and participation in the patient’s self-care. The trial’s successful results on outcomes during the nursing professional monthly interventions demonstrated a favourable evolution on outcome improvement over 18 months, with promising results on the first 6-month measures. The comparison between the trial CG and GI longitudinal results showed a positive evolution of HbA1c values from month 0 to month 18. It was observed that figures decreased in the IG and were maintained in the GC, with statistically significant differences (*p* ≤ 0.001). Levels achieved in the GI were very important (a reduction of 1.69%). We consider that, if these values were maintained, they would mean a positive impact for a better disease prognosis, expecting a decrease in the long-term micro- and macro-vascular complications to up to 42%, as concluded in the study by Johnson et al. [39]. Once the study was completed, the mean HbA1c values were lower than those obtained in prior studies with regard to the intervention group [40,41,42,43]; however, one of these studies was based on interventions with changes in pharmacological treatment and, in the other study, the intervention was implemented through telemedicine. In this regard, the systematic review by Almutairi et al. [44] identified telephone interventions that were culturally adapted to the population as the most effective, coinciding with the hereby presented results of the TELE-DD trial.

Behavioural-change based interventions have been attempting to increase treatment compliance for a long time, as reported in the Cochrane review of 83 studies by Haynes et al. [45]. However, the most effective interventions did not produce large improvements in treatment adherence [46], since the factors influencing treatment adherence behaviour are complex and unique to each individual, requiring numerous multifactorial strategies, some of them highly individualised, to remove barriers and promote adherence [47]. In our opinion, the optimal results obtained in the TELE-DD RCT are a consequence of influencing the physical and psychological aspects of lifestyle modification, and not just an adequate medication intake. The effect on adherence exerted by personal contact between the nurse and patient is the basis of the TELE-DD trial, noticing the importance of the human factor. The proximity and immediacy of patients being able to ask any questions during the monthly telephone call with the intervention nurses is a key aspect for the telephone intervention being more effective [48,49].

Changes in clinical outcomes were evident as early as 6 months after the trial onset. This crucial result suggests that the estimated time of improvement in the objectives of this RCT was shorter than initially considered and can serve as a guide if interventions of these characteristics are to be implemented in any population with chronic diseases, which are nonadherent to treatments (including lifestyle). This fact would have a positive impact on the reduction in health care costs. In this sense, it would be interesting to carry out a prognostic and cost-effectiveness study on the same patients who participated in the trial to check whether there is a maintenance of adherence and clinical status even years after the intervention period, or whether this is a temporary modification of behaviour while the intervention is continued by the nurses. The remaining biochemical and clinical outcomes such as LDL-cholesterol, blood pressure, and BMI also showed significant differences between the two groups, coinciding with the results of prior similar studies [50]. These further results would confirm prior findings from studies that associate condition prognosis improvement with treatment adherence and healthy habits through the control of cardiovascular risk factors [51].

The PHQ-9 depression questionnaire showed significant differences from month 6 and month 12 in the IG versus the CG (*p* ≤ 0.001), reaching values indicating “No depression”, which remained constant until 18 months, with results higher than those in the study by Osborn et al. [52]. A meta-analysis published by Manea et al. [53] indicated that: “if the algorithm scoring method is used, the PHQ-9 has a low sensitivity for detecting MDD. This could be due to the rating scale categories of the measure, higher specificity, or other factors that deserve further research”. The previously proposed PHQ-9 cutoff point of ≥10 presents a higher capability for diagnoses when screening, or where high sensitivity is needed. In this regard, and for the classification and diagnosis of patients with comorbid depression, recent data from Cichón et al. [54] recommend lower cutoff scores of the PHQ-9 for people with type 2 diabetes compared to the general population. It may be necessary to adjust these cutoff scores in these patients due to the high risk of depression in this pathology.

The results of this study highlight that DDS scores decreased in the intervention group over time. Comparing our data with the meta-analysis of Perrin et al. [55], the prevalence rate of diabetes distress was higher, but in patients with a higher prevalence of comorbid depression; the same results were obtained in the work of Martínez-Vega et al. [56] in the Mexican population. The data obtained were all evaluated according to the cutoff points of Fisher L et al. [57], which did not vary in the validations consulted in Spain [58]. When analysing the different DDS domain data on the trial sample, we found divergent results, but all findings indicated lower degrees of distress when compared with the samples of other consulted studies [59]. This could possibly be because public health care in Spain is universal and free of charge; diabetes care for all patients is guaranteed and treatments are fully covered, in some cases with a minimal contribution from the patients. In addition, an association was observed between the indicators of depression and emotional distress, where the lower the depression, the lower the degree of emotional distress, reflected in all domains of the DDS. A recent systematic review has shown that we can lower diabetes distress by using tailored interventions [60], although it may be difficult to separate some diabetes-specific distress from the usual depressive symptoms. We further suggest here the thoughts of Fisher, Gonzalez, and Polonsky [61]: “all patients, even those whose emotional distress rises to the level of MDD or anxiety disorders, may benefit from consideration of the content of distress to effectively targeted care, and suggest strategies for integrating the emotional side of diabetes”.

Limitations: The main limitations in the TELE-DD trial are that the MPR measuring method assumes that the medication collected at the pharmacy is always consumed by the patients and that the medication not collected at the pharmacy is considered as poor adherence. The change in dosage or medication for these pathologies could be considered as a limitation at the time of preparing the study, but it was considered that if the new dispensing was consumed by the patient correctly in time and form, it would be counted as good adherence. Another limitation of the RCT is to assess further adherence or non-adherence to treatment only using the MGB survey, since it is a self-administered questionnaire. However, the objective clinical data on improvement in all the parameters of the patients who said they were adherent coincided. Additionally, as mentioned in the discussion, lower PHQ-9 cutoff points for depression in patients with diabetes, should be used. Prior studies demonstrated that people with diabetes need a lower cutoff than others, specifically for depression screening with PHQ-9, but for the moment, the established ones were applied [54].

## 5. Conclusions

The TELE-DD intervention described and analysed in this report was based on the ADA recommendations to act on patients with T2D and comorbid clinical depression, particularly those with poor adherence to treatment. Good diabetes self-management and nurse personalised attention improve adherence, and optimal levels are achieved in clinical parameters while reducing depression symptoms and diabetes distress, which has an important impact on the quality of life of patients with T2D and comorbid depression. 

The TELE-DD team aims to translate the current study results to the wider health system population of those patients who do not finally retire their medication as prescribed in the pharmacy, according to the electronic prescription registry, and to develop clinical programs to improve adherence to treatment in this regard, in the context of the entire public health system.

## Figures and Tables

**Figure 1 ijerph-20-00328-f001:**
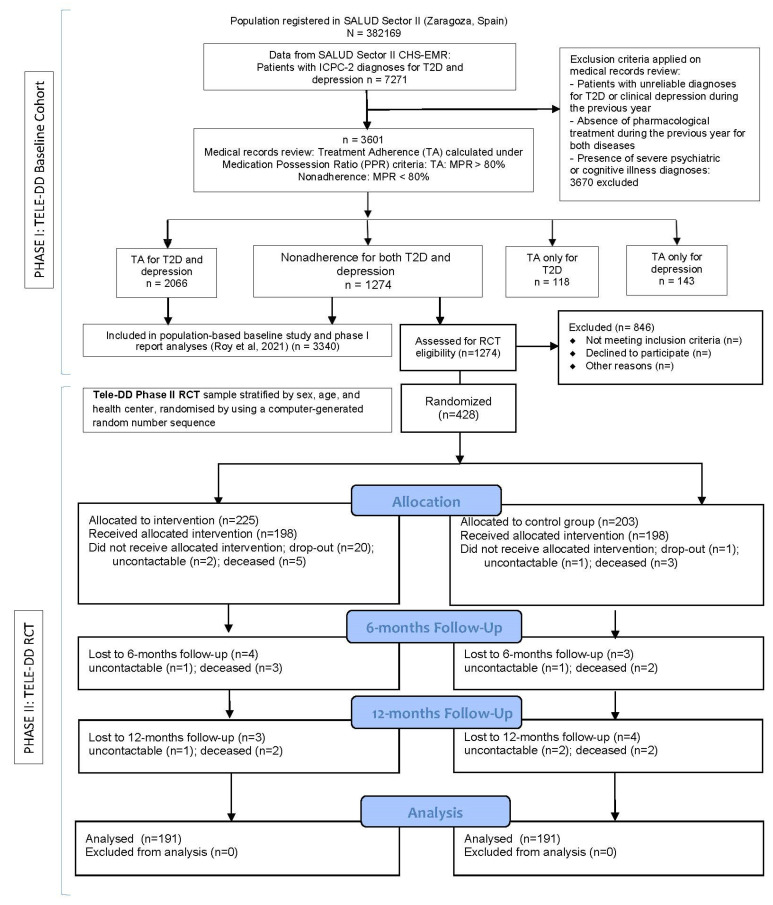
Flow diagram demonstrating the TELE-DD Project population data cohort (Phase I) and RCT sample randomisation, allocation, and follow-up (Phase II).

**Figure 2 ijerph-20-00328-f002:**
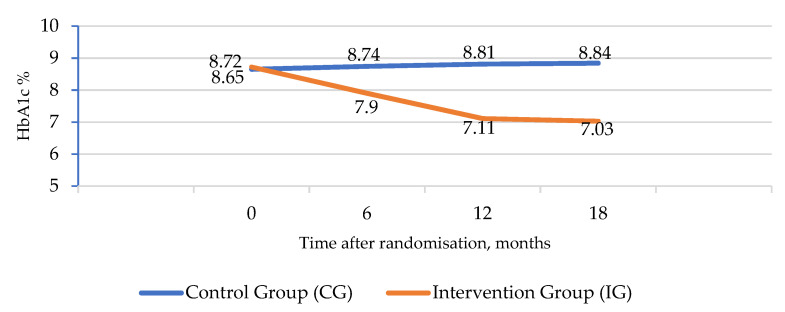
Evolution of HbA1c in the two groups of the RCT study (data extracted from Table 2).

**Figure 3 ijerph-20-00328-f003:**
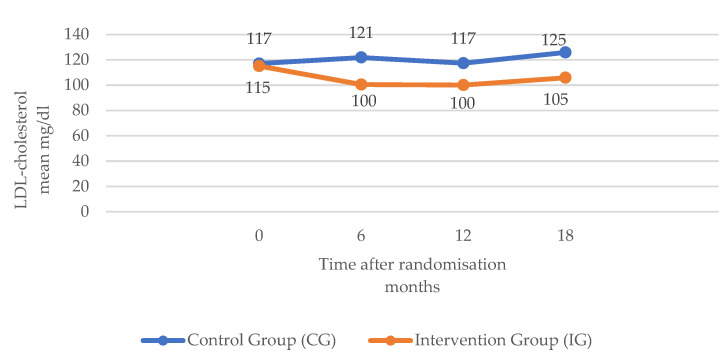
Evolution of (LDL) cholesterol in the two groups in the RCT study.

**Figure 4 ijerph-20-00328-f004:**
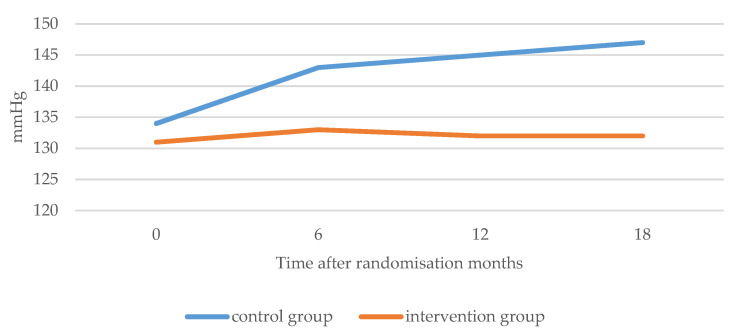
Evolution of systolic blood pressure in the two groups in the RCT study.

**Table 1 ijerph-20-00328-t001:** Sociodemographic and clinical data of the RCT sample (CG and IG) at the baseline (month 0).

Variable	RCT Group		*p*
	Control Group	Intervention Group	
Age *	71.5 (10.5)	71.4 (10.3)	0.857
Age IC. mean (95%)	70.1–73.0	70.0–72.7	
<50 years *	5 (2.5%)	8 (3.6%)	0.626
50–60 years *	31 (15.3%)	27 (12.0%)	
61–70 years *	48 (23.6%)	65 (2.9%)	
71–80 years *	76 (37.4%)	78 (34.7%)	
>80 years *	43 (21.2%)	47 (20.9%)	
Gender			
Male **	57 (28.1%)	64 (28.4%)	0.933
Female **	146 (71%)	161 (71.6%)	
Years of diagnoses T2D			
Mean (SD)	9.8 (5.5)	10.2 (5.7)	0.470
C. mean (95%)	9.0–10.6	9.4–10.9	
<5 years n (%)	35 (17.2%)	36 (16.0%)	0.831
5–10 years n (%)	80 (39.4%)	85 (37.8%)	
>10 years n (%)	88 (43.3%)	104 (46.2%)	
Years of diagnoses CD ***			
Mean (SD)	9.6 (4.8)	9.4 (5.3)	0.766
IC. mean (95%)	8.9–10.3	8.7–10.1	
<5 years n (%)	33 (16.3%)	44 (19.6%)	0.662
5–10 years n (%)	79 (38.9%)	86 (38.2%)	
>10 years n (%)	91 (44.8%)	95 (42.2%)	
Number of nurse consultation appointments			
mean (SD)	9.5 (10.4)	9.9 (11.7)	0.649
IC. Mean (95%)	8.0–10.9	8.4–11.5	
<5 inquiries **	74 (36.5%)	69 (30.7%)	0.098
5–10 inquiries **	61 (30.0%)	90 (40.0%)	
>10 inquiries **	68 (33.5%)	66 (29.3%)	
Number of Medical consultations			
mean (SD)	11.0 (6.7)	10.8 (6.7)	0.726
IC. mean (95%)	10.1–11.9	9.9–11.7	
<5 inquiries **	32 (15.8%)	34 (15.1%)	0.475
5–10 inquiries **	76 (37.4%)	97 (43.1%)	
>10 inquiries **	95 (46.8%)	94 (41.8%)	

* Mean (SD); ** n (%). T2D = diabetes mellitus; CD = clinical depression (n = 428). *t*-test and Chi-square.

**Table 2 ijerph-20-00328-t002:** Evolution of primary and secondary metabolic outcomes in the control and intervention groups during the RCT 18-months follow-up.

Variable	Control Group	Intervention Group	*p*
HbA1c (%)	Mean (SD)	Mean (SD)	
Month 0	8.65 (1.40)	8.72 (1.49)	<0.001 ^1^ <0.001 ^2^ <0.001 ^3^
Month 6	8.74 (1.40)	7.90 (1.22)	
Month 12	8.81 (1.38)	7.11 (1.17)	
Month 18	8.84 (1.38)	7.03 (1.09)	
LDL cholesterol (mg/dL)			
Month 0	117.07 (48.12)	115.09 (47.52)	<0.001 ^1^ <0.001 ^2^ <0.001 ^3^
Month 6	121.85 (43.29)	100.46 (34.85)	
Month 12	117.36 (40.03)	100.10 (38.11)	
Month 18	125.87 (38.60)	105.86 (35.43)	
Body mass index (BMI)			
Month 0	30.49 (5.11)	30.42 (5.04)	0.162 ^1^ 0.434 ^2^ <0.001 ^3^
Month 6	30.48 (5.12)	30.20 (4.99)	
Month 12	30.67 (5.09)	30.16 (5.10)	
Month 18	30.78 (5.02)	30.05 (5.09)	
Systolic Blood Pressure (mmHg)			
Month 0	134.41 (15.41)	131.83 (15.42)	<0.001 ^1^ <0.001 ^2^ <0.001 ^3^
Month 6	143.03 (15.36)	133.35 (14.35)	
Month 12	145.99 (12.13)	132.21 (15.26)	
Month 18	147.29 (11.20)	132.14 (10.82)	
Diastolic Blood Pressure (mmHg)			
Month 0	76.86 (9.17)	75.49 (9.60)	<0.001 ^1^ <0.013 ^2^ <0.001 ^3^
Month 6	75.13 (10.56)	74.84 (9.68)	
Month 12	75.20 (9.43)	73.11 (10.20)	
Month 18	74.25 (8.72)	70.25 (7.65)	

^1^ Time effect; ^2^ Group effect; ^3^ Interaction.

**Table 3 ijerph-20-00328-t003:** Evolution of the psychological outcomes in the RCT control group (CG) and intervention group (IG) during the 18-months follow-up.

	Control Group			Intervention Group			*p*
PHQ-Total *	** No Depression	*** Minor Depression	**** Major Depression	** No Depression	*** Minor Depression	**** Major Depression	
Month 0	75 (36.9%)	98 (48.3%)	30 (14.8%)	85 (37.8%)	108 (48.0%)	32 (14.2%)	0.978
Month 6	65 (32.8%)	108 (54.5%)	25 (12.6%)	103 (52.0%)	77 (38.9%)	18 (9.1%)	0.001
Month 12	78 (40.2%)	95 (49.0%)	21 (10.8%)	122 (62.6%)	62 (31.8%)	11 (5.6%)	<0.001
Month 18	71 (37.2%)	97 (50.8%)	23 (12.0%)	119 (62.3%)	69 (36.1%)	3 (1.6%)	<0.001
Chi-Square test							
DDS Total	Control Group ^0^			Intervention group ^0^			*p*
Month 0	1.50 (0.41)			1.48 (0.40)			<0.001 ^1^ <0.001 ^2^ <0.001 ^3^
Month 6	1.52 (0.42)			1.25 (0.31)			
Month 12	1.48 (0.46)			1.21 (0.30)			
Month 18	1.48 (0.50)			1.18 (0.27)			

* Results expressed n (%); ** No depression (PHQ-9 < 5); *** Minor depression (PHQ-9 = 5–14); **** Major depression (PHQ-9 ≥ 15). ^0^ Mean (standard deviation); ^1^ Time effect; ^2^ Group effect; ^3^ Interaction.

## Data Availability

We are working on including the data in the platform of the University of Zaragoza.

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
