# Peer review of "The TELE-DD Randomised Controlled Trial on Treatment Adherence in Patients with Type 2 Diabetes and Comorbid Depression: Clinical Outcomes after 18-Month Follow-Up"

_ijerph, 2022, doi:10.3390/ijerph20010328_

Round 1

Reviewer 1 Report

IJERPH (ISSN 1660-4601)

ijerph-2058085

Title: The TELE-DD randomised controlled trial on treatment adherence in patients with type2 diabetes and comorbid depression: Clinical outcomes after 18-month follow-up

Comments:

1) What is the Role of oral hypoglycemic agents and antidepressant medications, glycemic control, and depression among primary care patients

2) what about the success rate of using glycated hemoglobin (HbA(1c)) assays to measure glycemic control and the 9-item Patient Health Questionnaire (PHQ-9) to assess depression, in reference to Medication Event Monitoring system? 

3)Does integrated approach to depression and type 2 diabetes treatment may facilitate its deployment in real-world practices with competing demands for limited resources? 

4) If  DM is undiagnosed or untreated, it will significantly increase cardiovascular risk and reduce life expectancy while increasing the risk of hospital admissions, and health costs. It needs correlation with sufficient data.

5) Need sufficient data interpretation to determine the point prevalence of T2D and clinical depression comorbidity and treatment nonadherence; (2) to test if HbA1c and LDL-C, as primary DM outcomes, are related to TA.

Reviewer 2 Report

1. I would like to see your raw data and the details of your telephone follow-up visit

2. In your experimental design, you designed a control group and an intervention group, and you followed up the intervention group with a telephone call, etc. In the results, you showed an improvement in the intervention group, but how can you tell whether this improvement was due to the content of the telephone call or to the respondents' expectations of the telephone call?

Author Response

As for the database, I am sorry but I am having problems. I am trying to send you the database but as I don't have the SPSS software license installed at the moment, it does not allow me to do it.

Reviewer 3 Report

Please see the pdf file with few comments
